# Neural correlates of the uncanny valley effect for robots and hyper-realistic masks

Shona Fitzpatrick[1], Ailish K. Byrne [2], Alex Headley[1], Jet G. Sanders[3], Helen Petrie[4], Rob Jenkins[1], Daniel H. Baker[1]*

1 Department of Psychology, University of York, York, United Kingdom, 2 School of Medicine, Keele University, Newcastle-under-Lyme, Staffordshire, United Kingdom, 3 Department of Psychological and Behavioural Science, London School of Economics, London, United Kingdom, 4 Department of Computer Science, University of York, York, United Kingdom

* daniel.baker@york.ac.uk

## Abstract

Viewing artificial objects and images that are designed to appear human can elicit a sense of unease, referred to as the 'uncanny valley' effect. Here we investigate neural correlates of the uncanny valley, using still images of androids (robots designed to look human), and humans wearing hyper-realistic silicone masks, as well as still images of real humans, in two experiments. In both experiments, human-like stimuli were harder to distinguish from real human faces than stimuli that were clearly not designed to mimic humans but contain facial features (mechanical robots and Halloween masks). Stimulus evoked potentials (electromagnetic brain responses) did not show convincing differences between faces and either androids or realistic masks when using traditional univariate statistical tests. However, a more sensitive multivariate analysis identified two regions of above-chance decoding, indicating neural differences in the response between human faces and androids/realistic masks. The first time window was around 100–200 ms post stimulus onset, and most likely corresponds to low-level image differences between conditions. The second time window was around 600 ms post stimulus onset, and may reflect top-down processing, and may correspond to the subjective sense of unease characteristic of the uncanny valley effect. Objective neural components might be used in future to rapidly train generative artificial intelligence systems to produce more realistic images that are perceived as natural by human observers.

## Introduction

Many people report an aversion to entities that are superficially human-like, but on closer inspection turn out to be artificial. Examples include humanoid robots (androids), puppets, hyper-realistic masks [1], and computer-generated images or movies. The term 'uncanny valley' [2] (English translation in [3]) describes the idea that clearly human or clearly artificial entities do not evoke unease, whereas artificial entities that are human-like are disconcerting. Understanding these experiences is increasingly important as artificial entities become more integrated into our everyday lives. However at present relatively little is known about the neural underpinnings of the uncanny valley effect (for a recent review, see [4]). In particular, the

**Data availability statement:** Raw data, processed data, and analysis scripts are freely available through the project repository at: https://doi.org/10.17605/OSF.IO/5NZ2H.

**Funding:** SF was funded by a YorRobots Venables internship. JGS was funded by the Economic and Social Research Council (Studentship ES/J500215/1) and Prins Bernhard Cultuurfonds (grant E/30/30.13.0630/HVH/IE). Also supported by BBSRC grant BB/V007580/1 awarded to DHB.

**Competing interests:** The authors have declared that no competing interests exist.

root of the uncanny valley effect remains debated: does it arise primarily from bottom-up sensory conflicts, or from higher-level cognitive processes? Resolving this question is critical to understanding its fundamental mechanisms.

Neural responses to faces and bodies in general are well-characterised, and there appear to be specialised brain regions devoted to both (reviewed in [5]). For example, areas of the occipital lobe [6] and fusiform gyrus [7] respond more to faces than to non-face stimuli, and sections of extrastriate cortex are responsive to bodies [8]. There are also electromagnetic event-related potential (ERP) signals associated with face and body stimuli, though their precise role is still debated [9,10]. It seems highly likely that 'uncanny' images will activate these same processes, yet it is unclear whether the initial cause of the sense of unease they produce occurs at bottom-up sensory stages [11–13] or is modulated by more top-down cognitive factors [14,15].

One previous study by [16] measured functional magnetic resonance imaging (fMRI) responses to moving stimuli designed to elicit an uncanny valley effect. They found repetition suppression effects (repetition suppression is a phenomenon in which the neural response to repeated presentations of identical or similar stimuli is reduced relative to the response on the first presentation) in action-specific brain regions responding to movies of androids that had a biological appearance, but mechanical motion. These effects were stronger than for movies of humans or mechanical robots performing the same actions. A more recent electroencephalography (EEG) study [17] identified a difference in the N400 component (the N400 is an electromagnetic brain potential obtained 400 ms after stimulus onset, typically over centro-parietal electrodes; it has been proposed to reflect the extent to which the stimulus presented was surprising or unexpected) between dynamic and static conditions using the same stimuli. Although this difference was strongest over frontal electrodes, source reconstruction of the N400 itself suggested a left-lateralised source in the temporo-parietal cortex, consistent with the fMRI results [16]. The authors interpret both of these findings as being due to the discrepancy between the human-like appearance and the clearly non-biological motion of the robot.

Our aim was to further investigate neural correlates of the uncanny valley effect, with the expectation that increased understanding will aid efforts to generate more convincingly human robots and avatars in the future. We achieved this through two EEG experiments, in which we measured neural responses to static images. Although previous studies focus on dynamic stimuli, static images allow for a more precise investigation of the neural mechanisms underlying the uncanny valley effect, particularly by eliminating motion-related confounds. In the first experiment, the stimuli were still images of humans, machine-like robots, and human-like robots (see Fig 1a). In the second experiment we aimed to generalise the finding by using images of people wearing no masks, wearing obvious masks (e.g. Halloween masks), and wearing hyper-realistic silicone masks [18] (see Fig 1b). Rather than focus on specific ERP components, we use a multivariate pattern classification approach (a machine learning technique in which an algorithm is trained to decode the neural responses) to identify time windows in which information in the EEG signal can be used to distinguish between pairs of conditions. Our rationale is that timepoints where signals evoked by human faces can be distinguished from those evoked by human-like robots, or hyper-realistic masks, are candidates for a neural signature of the uncanny valley effect.

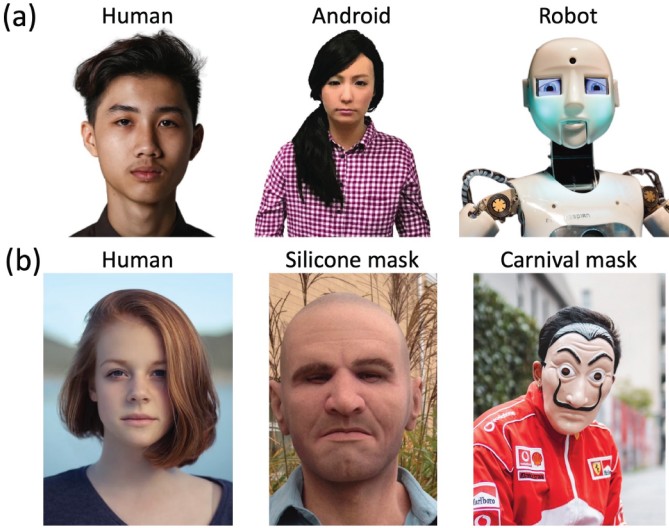

**Fig 1. Illustrative stimuli from the same categories as used in Experiments 1 and 2.** Row (a) shows a human face, an android and a robot, all against white backgrounds. Row (b) shows a human face, a hyper-realistic silicone mask, and a Halloween mask, all against natural backgrounds. Images shown here were taken from a variety of sources that permit reuse in academic contexts and in most cases were not part of the stimulus set from the experiments. The silicone mask image was taken by the authors (subject: RJ, image credit: JGS), and was used in Experiment 2. The individual in this manuscript has given written informed consent (as outlined in PLOS consent form) to publish these case details (i.e. this image).

## Materials and methods

### Participants

A total of 29 participants completed Experiment 1 (12 male, 17 female), and 30 different participants completed Experiment 2 (7 male, 23 female). Participants were young adults with no history of neurological disorder. None of the participants had previously taken part in a study using these stimuli, and all were naïve to the hypotheses and wore their normal optical correction if required. Written informed consent was collected before each experiment began, and all procedures were approved by the Ethics committee of the Department of Psychology at the University of York. Data collection for Experiment 1 ran from 14th July to 15th September 2022, and data collection for Experiment 2 ran from 12th October 2017 to 14th February 2018.

### Apparatus and stimuli

In Experiment 1, the stimulus set consisted of a total of 90 images, evenly split between three categories: real faces, human-like robots, and mechanical robots. Images all showed the head and shoulders of the subject, had white backgrounds, and were sourced from the Internet. In Experiment 2, the stimulus set (first described by [19], but here including additional images) consisted of a total of 296 images, comprising real faces (148 images), people wearing silicone masks (74 images), and people wearing obvious masks of the sort typically worn for carnivals and Halloween celebrations (74 images). The backgrounds of these images were more heterogeneous, and showed the natural surroundings of the subject. While the image backgrounds differed across experiments, we hypothesize that the primary task was not affected,

as participants focused on the foreground stimuli. In both experiments, images included examples of both genders, and of varied ethnic backgrounds.

All stimuli were displayed on a ViewPixx display running at 120 Hz, controlled by an Apple Macintosh computer. The display was gamma corrected using a photometer to ensure that the luminance output was linear. EEG data were collected using a 64-channel Waveguard cap and an ANT Neuroscan system, sampling at 1 kHz. The ground electrode was located at position *AFz*, and all signals were referenced to the whole head average. Low latency digital triggers were sent between the display and the EEG amplifier using an 8-bit parallel cable.

## Procedure

**Experiment 1: Robots.** Each participant completed three blocks of the first experiment. Within each block, all 90 stimulus images were presented twice in a random order. Stimuli subtended 11 × 11 degrees at the viewing distance of 57 cm, and were shown against a mid-grey background, with a black central fixation cross displayed throughout. The presentation duration was 500 ms, and participants were asked to press a mouse button to indicate if they believed each image was of a human or of a robot/android. After each response there was a random duration blank period with a mean duration of 1000 ms and a standard deviation of 200 ms. Durations were chosen to provide sufficient information for judgment while avoiding task fatigue. Randomized blank periods were designed to reduce carryover effects and prevent anticipatory biases. Each block lasted around 6 minutes.

After the EEG experiment, participants also completed a series of questionnaires using the Qualtrics platform. These involved rating their perception of a subset of the stimuli (8 from each category), using items from the Godspeed questionnaire [20]. Items were selected that were expected to be most closely aligned to measuring the sensation of uncanniness. Inspection time was unlimited. Participants also provided demographic information (age, gender) and completed the GAToRS [21] and AQ [22] questionnaires, however the results of these additional questionnaires are not presented here.

**Experiment 2: Hyper-realistic masks.** Participants were shown all 296 images in a random order in each of three blocks. In the first block, stimuli subtended 5.5 × 7.5 degrees of visual angle when viewed at a distance of 57 cm. In the second block, stimuli doubled in size (width and height), and subtended 11 × 15 degrees at the same viewing distance. In the third block, stimuli doubled in size again, and subtended 22 × 30 degrees. The rationale for the size manipulation was to investigate whether increasing levels of detail made the silicone masks more identifiable [18,19]. However as that is not the main focus of the current paper, and our preliminary analyses indicated no differences between size conditions, we collapse results across size conditions. Stimuli were presented for 250 ms, and participants indicated whether they thought each image contained a real face or a mask, using a two-button trackball. The button assignment (whether the left button indicated a face or a mask, and vice versa) was determined randomly for each participant, but remained constant throughout the whole experiment. Text reminding the participant of the button assignment was present continuously in the lower right corner of the screen, far from the area of the screen where the stimuli were presented. A central fixation cross was also present throughout. After each response there was a random duration blank period with a mean duration of 1000 ms and a standard deviation of 200 ms. Each block lasted around 8 minutes.

An independent group of 20 participants also completed an online questionnaire in which they rated the images along various dimensions. The participants repeated the real face vs mask judgement from the main experiment, and were additionally asked to rate emotional

expressiveness, realism, and uncanniness for each image using a 7-point Likert scale. Inspection time was unlimited for these judgments.

## Data analysis

We analysed response data by calculating d-prime ($d'$) scores for each condition, derived from the hit rate and false alarm rate [23]. For the human conditions, the hit rate was the proportion of human images correctly identified as human, and the false alarm rate was the proportion of robot or mask images that were incorrectly judged as being human. For the robot and mask conditions, the hit rate was the proportion of robot/mask images correctly identified as not being human, and the false alarm rate was the proportion of human images that were incorrectly judged as being non-human (note that this means the false alarm rate was the same for the robot and android conditions, and for the silicone and Halloween mask conditions). We capped infinite d-prime values (which occur e.g. when the hit rate is 1) at an arbitrary ceiling of 5 to prevent outliers from skewing the results, following established conventions in signal detection theory. We log transformed the reaction times (which typically have positive skew) and performed all averaging and statistical analysis on the logarithmic values.

EEG signals were recorded during each block and saved to disc for subsequent offline analysis. We used components of the EEGlab toolbox [24] to import the data into Matlab and collate data across blocks. We then used Brainstorm [25] to filter the data using a bandpass filter (0.5 to 30 Hz), epoch by condition, and subtract a pre-trial baseline (the mean voltage for the 200 ms before stimulus onset). Five participants were excluded from the EEG analysis of each experiment due to excessive noise. Our attempts to clean up the data from these participants using independent components analysis were unsuccessful. Their behavioural data were unaffected, and are therefore still included in the analysis.

We performed univariate analyses by conducting Bayesian t-tests [26] between ERPs from pairs of conditions at each time point using a JZS prior, using signals pooled across electrodes P6, P8, PO6 and PO8, which are typically associated with visual responses to faces. The resulting Bayes factor score is a summary of the evidence in favour of either the null hypothesis (that the waveforms are equal) or the alternative hypothesis (that they differ). We use the heuristics proposed by Jeffreys [27] that Bayes factors >3 ($log_{10}BF_{10} > 0.5$) constitute some evidence supporting the alternative hypothesis, factors >10 ($log_{10}BF_{10} > 1$) constitute strong evidence, and factors >30 ($log_{10}BF_{10} > 1.5$) constitute very strong evidence.

Multivariate pattern analysis (MVPA) was conducted by training a linear support vector machine algorithm (LibSVM, [28]) to discriminate between patterns of activity across electrodes at a specific time point. MVPA is a statistical technique that involves training a machine learning algorithm to identify patterns in data, and then testing its accuracy at classifying unseen data; in EEG analysis above-chance classification is considered evidence of distinct patterns of neural activity between two conditions. Previous work has indicated that EEG data do not typically require more complex nonlinear algorithms [29]. The patterns came from the human face condition and one of the other conditions, for a single participant. Four examples of each pattern were calculated by averaging over random subsets of 20% of the available trials from a given condition, and these were used to train the classifier. The accuracy of the classifier was tested on the remaining trials (that were not used in training) for each condition. This process was repeated 1000 times with different trial permutations to obtain an average accuracy, where chance performance is at 50% correct. The analysis was carried out at all time points, and for each participant separately. We then averaged classifier accuracy across participants, and calculated one sample Bayesian t-tests comparing to chance performance at each time point as described above.

### Data and code availability

Raw data, processed data, and analysis scripts are freely available through the project repository at: https://doi.org/10.17605/OSF.IO/5NZ2H

## Results

### Experiment 1

We first explored the behavioural results for identification of human versus non-human stimuli. We calculated d-prime scores to compare sensitivity across conditions. Sensitivity was highest for identifying robots ($d'$ = 4), but still well above chance for both the human ($d'$ = 2.72) and android ($d'$ = 2.27) conditions. The Bayes factor score for a one-way ANOVA comparing these three conditions indicated very substantial evidence ($log_{10}BF_{10}$ = 6.67) for a difference between conditions, as illustrated in Fig 2a. Pairwise Bayesian t-tests between conditions indicate very convincing differences in sensitivity between robots and androids ($log_{10}BF_{10}$ = 7.15) and robots and humans ($log_{10}BF_{10}$ = 6.27). The difference between androids and humans ($log_{10}BF_{10}$ = 5.58) was also very substantial. The higher d' values for robots (Fig 2a) could indicate that these stimuli are more visually salient.

Reaction times also differed between conditions, though the effects were rather smaller. Reactions were fastest for identifying robots (RT = 688 ms), compared with humans (RT = 803 ms) and androids (RT = 809 ms). The Bayes factor score for a one-way ANOVA comparing these three conditions indicated strong evidence ($log_{10}BF_{10}$ = 1.44) for a difference between conditions, as illustrated in Fig 2b. Pairwise Bayesian t-tests between conditions indicate very convincing differences in sensitivity between robots and androids ($log_{10}BF_{10}$ = 5.01) and robots and humans ($log_{10}BF_{10}$ = 4.61), whereas the reaction time was equivalent between androids and humans ($log_{10}BF_{10}$ = –0.68).

EEG activity showed a clear visually evoked potential over posterior electrode sites (see Fig 2c), with typical components found in response to visual stimuli pooled across all conditions (the P100, N170 and P200 are indicated in the figure). Pairwise comparisons of conditions are shown in Fig 3a and 3b. In general there is a tendency for the ERP response to human faces to diverge slightly from the other two conditions [30], however the evidence for this divergence was not compelling. Bayes factors exceeded 3 for only a small number of time points around 300–400 ms in the comparison between human and robot images (see yellow bars at y = –8 in Fig 3a), but these differences were small considering the variance in the data.

We also conducted multivariate pattern analysis independently at each time-point for the same two comparisons. The evoked responses for human and robot images caused sufficiently distinct patterns of voltages across the scalp that the pattern classifier could distinguish between them from around 100 ms following stimulus onset, with accuracy up to 72% correct (see Fig 3c). Bayes factors exceeded 30 for much of the time window between 100 and 800 ms, indicating that the decoding was meaningfully above chance performance (50% correct). It was also possible to classify between human and android images (see Fig 3d), however performance was much poorer, with a maximum of 62% correct. Classification accuracy had an initial peak around 100 ms that provided compelling evidence for above chance classification (BF>30), and a later region of above-chance classification between 500 and 700 ms. The early time window likely reflects rapid processing of low-level visual features, such as edges or color contrasts, consistent with P100 and N170 components, whereas the later time window may involve higher-level cognitive processes, such as evaluating emotional content or judging authenticity. In the Discussion we speculate that these two time periods might correspond to distinct types of signal associated with the uncanny valley. More generally, the high

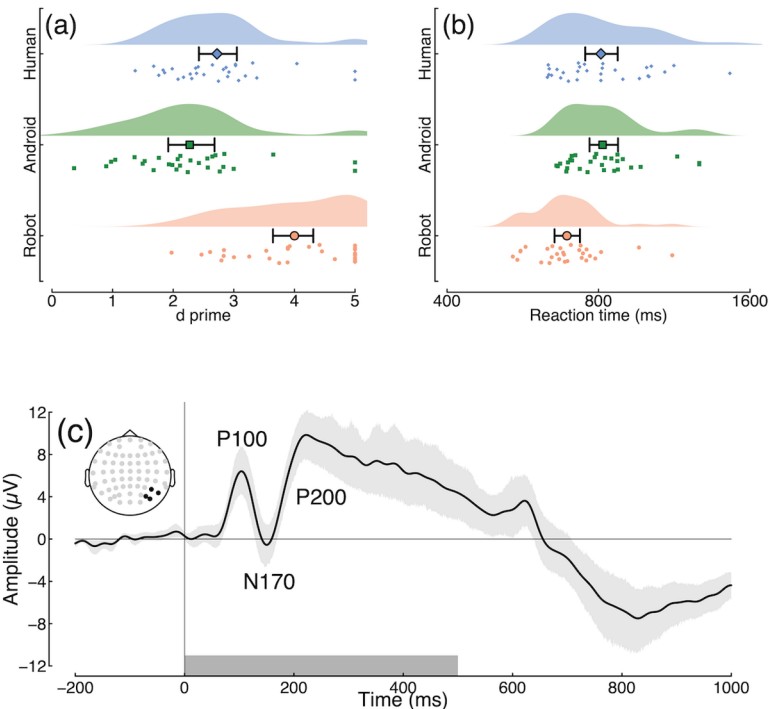

**Fig 2. Summary of response data and grand mean ERP for Experiment 1.** Panel (a) shows d-prime scores for identifying images of human (blue diamonds), android (green squares) and robot (red circles) faces. Small points show individual participants, and the larger symbols with error bars indicate the group mean and bootstrapped 95% confidence intervals. Panel (b) plots reaction times in the same format (note the logarithmic x-axis). Panel (c) shows the grand mean ERP across all participants and conditions, pooled across electrodes P6, P8, PO6 and PO8 (see inset). The P100 is a positive evoked potential occurring around 100 ms after onset of a visual stimulus, associated with the initial (low level) visual response; the N170 is a negative potential at 170 ms that is often associated with faces; the P200 is a further positive potential linked to attention and stimulus discrimination. The shaded region around the curve illustrates the 95% confidence interval, and the grey rectangle at the foot indicates the stimulus duration.

classification accuracy for human vs. robot stimuli may be attributed to the salient mechanical elements of robots, whereas the lower accuracy for humanoids reflects their ambiguous human-like appearance, leading to confusion.

Finally, we analysed the rating data from a set of 10 questionnaire items for 8 stimuli from each category. The results are summarised in Fig 4, and in general show differences between stimulus categories along most dimensions. Of particular note, the U-shaped function predicted by the uncanny valley effect was apparent for ratings along the Dislike-Like (Fig 4d), Unfriendly-Friendly (Fig 4e), and Anxious-Relaxed (Fig 4i) dimensions. These are all dimensions with emotional valence, indicating support for the 'uncanniness' of our android stimuli. However we note that the android and robot categories were typically rated as being more similar to each other than to the real human faces. Following Experiment 1, we sought to generalise our results to a different stimulus set, and next report the results of Experiment 2 which used hyper-realistic silicone masks.

## Experiment 2

The results of Experiment 2 were similar to those of Experiment 1, despite using a quite different stimulus set involving images of humans wearing masks, rather than robots. Sensitivity

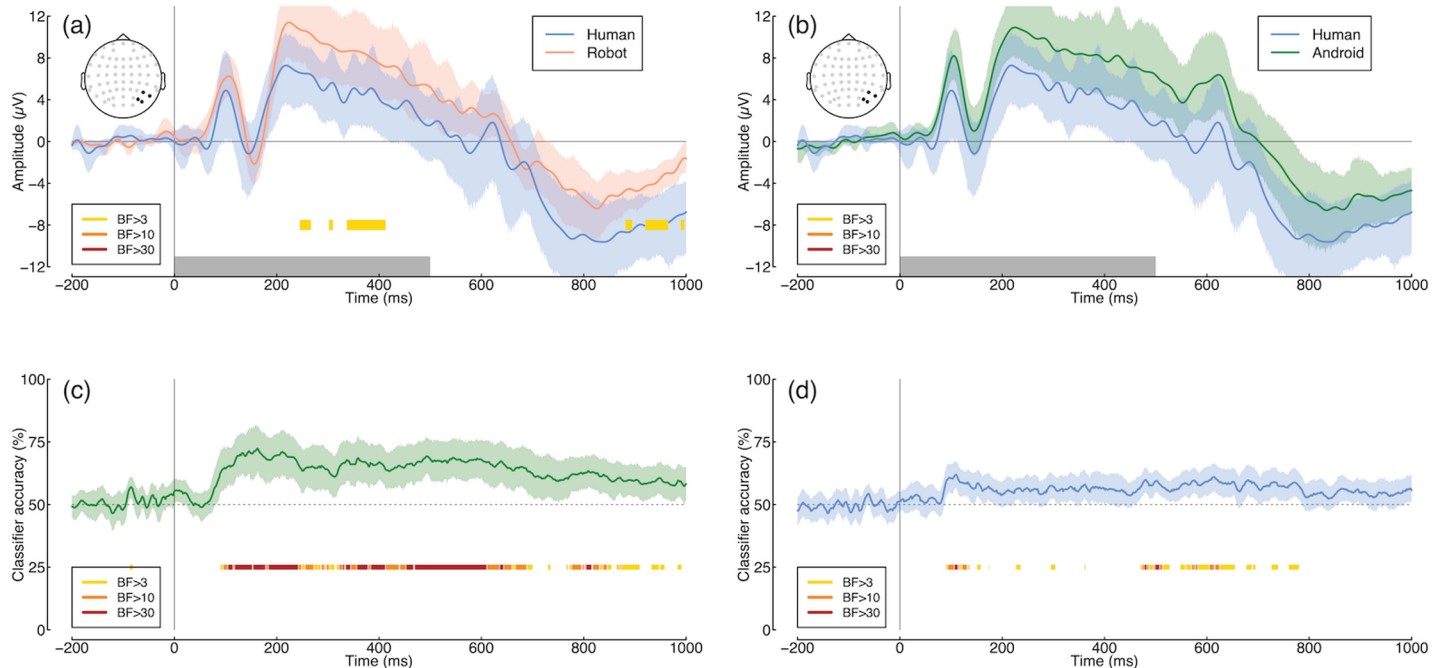

**Fig 3. Univariate and multivariate comparisons across image type**. Panel (a) shows the ERPs comparing human (blue) and robot (red) face images, and panel (b) compares human (blue) and android (green) faces. Panels (c) and (d) show multivariate pattern classification accuracy for the same comparisons. Points at y = −8 and y = 25 indicate Bayes factor scores for comparisons between ERPs (a,b) and comparing classification accuracy to chance (50% correct; c,d).

was highest for identifying Halloween masks ($d'$ = 4.01), but still well above chance for both the human ($d'$ = 2.52) and silicone mask ($d'$ = 1.29) conditions. The Bayes factor score for a one-way ANOVA comparing these three conditions indicated very substantial evidence ($log_{10}BF_{10}$ = 13.34) for a difference between conditions, as illustrated in Fig 5a. Pairwise Bayesian t-tests between conditions indicate very convincing differences in sensitivity between Halloween and silicone masks ($log_{10}BF_{10}$ = 9.71) between Halloween masks and humans ($log_{10}BF_{10}$ = 11.59), and between silicone masks and humans ($log_{10}BF_{10}$ = 5.19). Unlike in Experiment 1, there were no convincing reaction time differences between conditions ($log_{10}BF_{10}$ = –0.8), as illustrated in Fig 5b.

The grand average ERP waveform for Experiment 2 (see Fig 5c) had similar initial components as for Experiment 1. The latter portion of the waveforms differed somewhat, most likely owing to the difference in presentation duration across experiments (250 ms versus 500 ms). There was a substantial univariate difference in ERP response between human and Halloween mask conditions extending from around 170 to 230 ms following stimulus onset (see Fig 6a), with Bayes factors exceeding 30. Univariate differences between the human and silicone mask conditions were not compelling (see Fig 6b).

Multivariate pattern analysis revealed extremely high classification accuracy (up to 97% correct) comparing human faces with Halloween masks. This was convincingly above chance, with a Bayes factor score exceeding 30 from around 100 ms following stimulus onset, and extending across the full time window (see Fig 6c). Classification was also convincingly above chance when comparing human faces with silicone masks (Fig 6d). This timecourse had an initial peak of high accuracy (up to 76% correct) between 100 and 200 ms after stimulus onset, followed by a second peak around 600 ms. This replicates the finding from Experiment

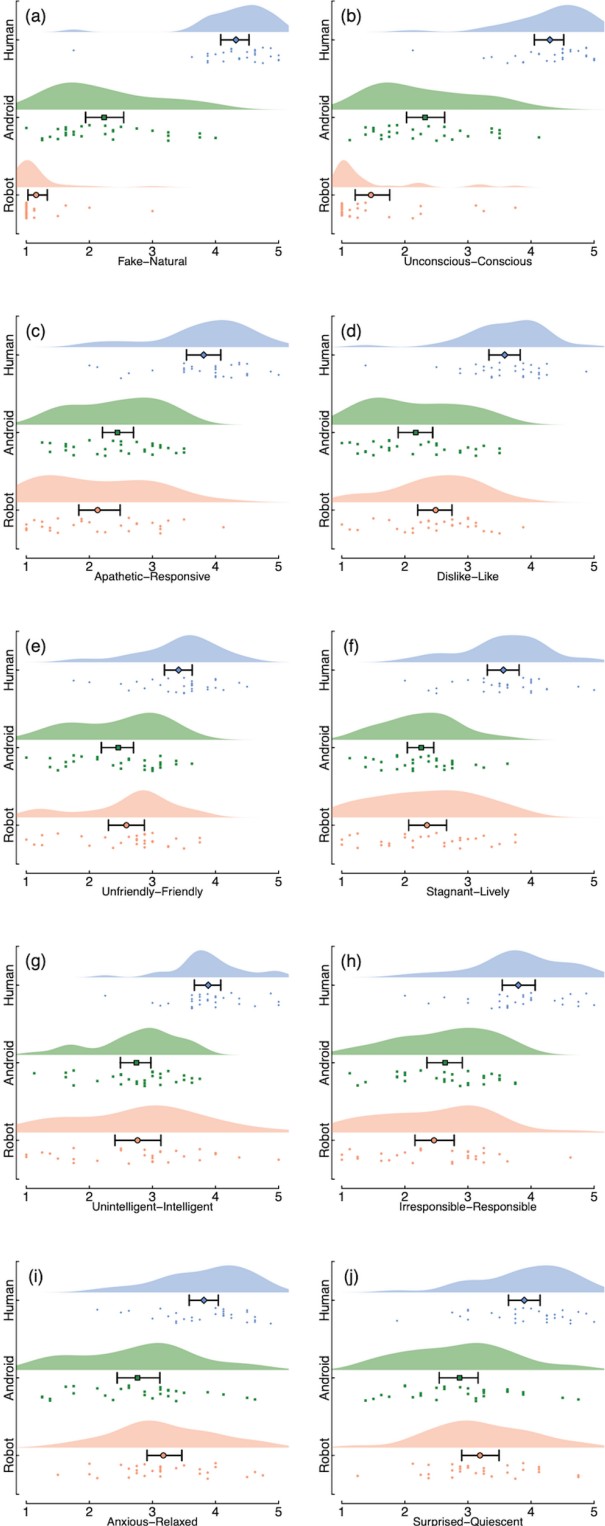

**Fig 4. Ratings of stimuli using items from the Godspeed questionnaire.** Each rating was on a Likert scale from 1–5, and was the average of ratings from 8 stimulus examples. Dots show individual participant scores, with the larger symbols indicating the mean and 95% bootstrapped confidence intervals.

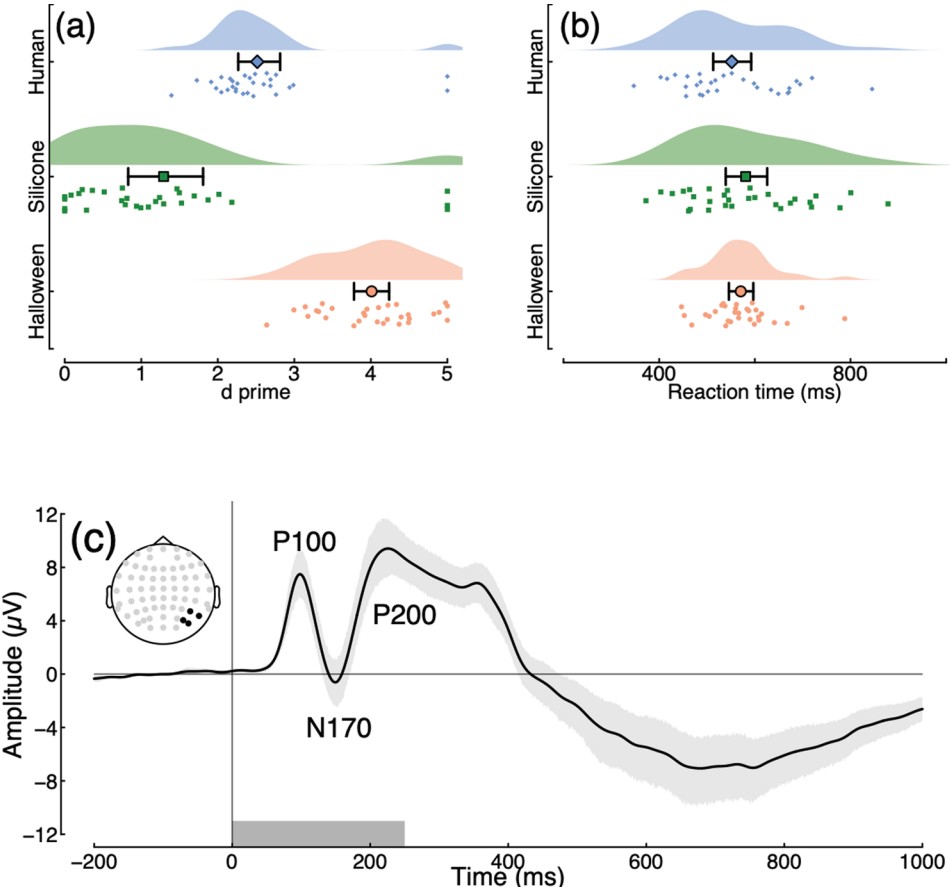

**Fig 5. Summary of response data and grand mean ERP for Experiment 2.** Panel (a) shows d-prime scores for identifying images of human faces (blue diamonds), silicone masks (green squares) and Halloween masks (red circles). Small points show individual participants, and the larger symbols with error bars indicate the group mean and bootstrapped 95% confidence intervals. Panel (b) plots reaction times in the same format (note the logarithmic x-axis). Panel (c) shows the grand mean ERP across all participants and conditions, pooled across electrodes P6, P8, PO6 and PO8 (see inset). The shaded region around the curve illustrates the 95% confidence interval, and the grey rectangle at the foot indicates the stimulus duration.

1 that uncanny valley responses might involve two distinct components at different moments in time.

We subsequently obtained ratings from an independent sample of 20 participants using the same stimuli as in the EEG experiment. This time, we asked for explicit ratings of emotional expressiveness, realism, and uncanniness, as well as repeating the binary real face vs mask rating. Real faces were rated highest for emotional expressiveness (M = 3.9) and realism (M = 5.9), and lowest for uncanniness (M = 2.7) (Fig 7a–7c). The realistic silicone masks were rated highest for uncanniness (M = 4.5), however this was not dramatically higher than for the Halloween masks (M = 4.3). Arguably making judgements of uncanniness is less appropriate for masks that are not intended to be realistic, though our data do qualitatively conform to the U-shaped function expected by the uncanny valley hypothesis. The pattern of d-prime scores (Fig 7d) was similar to those obtained in the main experiment (Fig 5b), with generally higher scores attributable to the unlimited inspection time permitted in this online follow-up experiment.

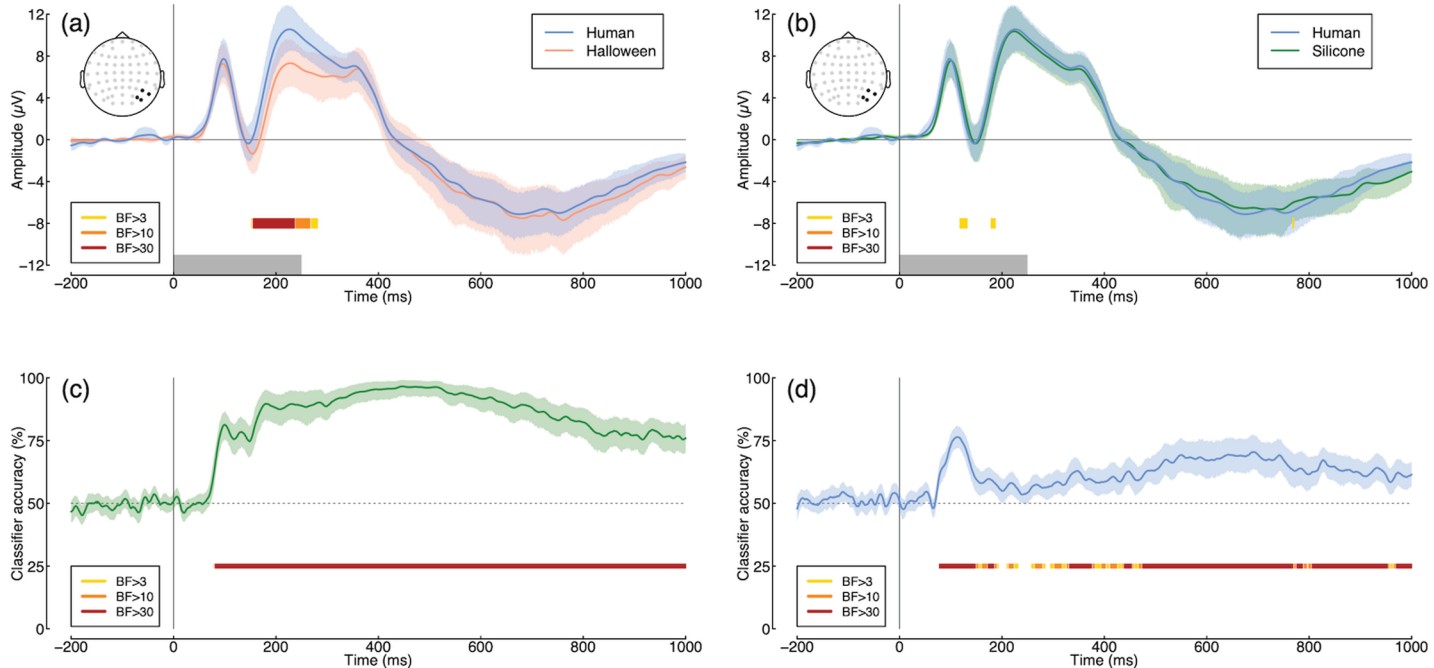

**Fig 6. Univariate and multivariate comparisons across image type for Experiment 2.** Panel (a) shows the ERPs comparing human faces (blue) and Halloween masks (red), and panel (b) compares human faces (blue) and silicone masks (green). Panels (c) and (d) show multivariate pattern classification accuracy for the same comparisons. Points at y = −8 and y = 25 indicate Bayes factor scores for comparisons between ERPs (a,b) and comparing classification accuracy to chance (50% correct; c,d).

## Discussion

Across two experiments using diverse stimuli, we identified a potential neurophysiological signature of the 'uncanny valley' effect. EEG responses to androids or silicone masks could be distinguished from responses to human faces at around 100 ms after stimulus onset, and also in a later time window around 500–800 ms after stimulus onset. There were no clear differences in the unimodal ERP response at posterior electrodes, but performance of a multivariate pattern classifier was above chance in these time windows. This is a different pattern from that observed for more obviously non-human stimuli (robots and Halloween masks), where there were both univariate and multivariate differences, and the multivariate discrimination accuracy was above chance for an extended time window. Perceptual judgements indicated that identification performance for uncanny valley stimuli was relatively poor, indicating confusion with real human images. We also confirmed that android images were perceived more negatively than either human or robot images, and that silicone masks were perceived as more uncanny than human faces. The similarity in results across our two experiments is striking and constitutes an internal conceptual replication of our main findings, suggesting that the neural characteristics of the uncanny valley effect are generalizable across stimulus categories.

The early time window when pattern classification is above chance corresponds approximately to the P100 and N170 components of the ERP. The P100 is typically associated with low-level visual responses, and is affected by contrast and spatial frequency content of an image. The N170 component is most often associated with faces, though is also observed for other image categories, and there is still debate about its precise function [9,10]. Similar early components have also been investigated in other ERP studies on the uncanny valley

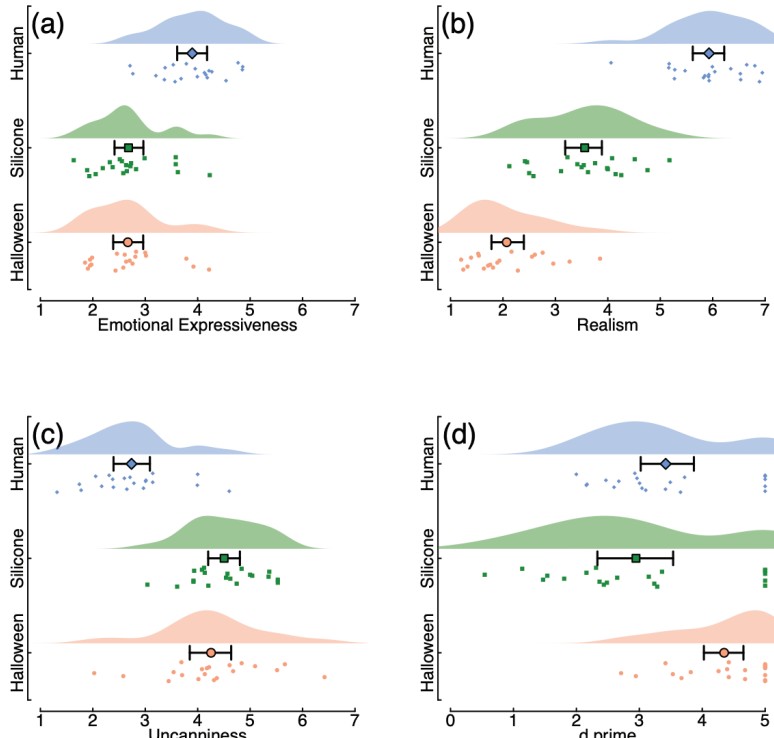

**Fig 7. Additional ratings of mask images, completed by an independent sample (N = 20) with unlimited inspection time.** Images were rated on three dimensions using seven-point scales (panels a–c), and also judged as being either a real face or a mask, from which d-prime scores were calculated (panel d).

effect [31,32], and comparing human and robot faces [30]. This time window is unlikely to be modulated substantially by top-down influences, so we attribute the early component to image-based differences between stimulus categories [33]. ERP components in later time windows have also been studied in previous work [17,34,35], and may reflect cognitive processing stages, such as determining whether a stimulus conforms to categorical expectations. Differences between stimulus categories at these times are more amenable to top-down influences, and most likely involve higher brain areas outside of occipital cortex. We therefore predict that ERP components at later time points should correspond with perceptual judgements and reports of uncanniness - this is a worthwhile hypothesis for future work to investigate.

Our use of hyper-realistic silicone masks is novel in the context of the study of uncanny valley effects. Previous studies using these masks have demonstrated that they are difficult to distinguish from real faces [18,19], including in applied settings such as simulated passport control [36,37], and show large individual differences [38]. Image analysis indicates that good identification performance for silicone masks is typically based on attention to the region below the eyes [38], however it is plausible that many observers are not explicitly aware of the cues they use to perform this judgement. This might contribute to both the early and late components identified in this study, and presumably also to the subjective sensation of 'uncanniness' that is characteristic of the phenomenon.

Another increasingly common situation that triggers the 'uncanny valley' experience is in the domain of computer-generated images and movies [39,40]. Artificial intelligence algorithms are now able to generate images and movies based on text prompts (for example

"a picture of a girl flying a kite in a field") that often include human subjects. However, at time of writing, images of humans often contain errors, such as the presence of too many limbs, digits, teeth etc. Synthetic movies often contain continuity errors, and have issues reproducing biological motion. Many of these errors are subtle and take time to spot, but it is also the case that human observers can report that images look 'wrong' without explicitly knowing why. The neural uncanny valley effect that we report here might prove a useful index of these instinctive reactions, and could even potentially be used to improve artificial intelligence algorithms. For example, images could be penalised for producing neural responses that differed from those for natural images.

More generally, the advantage of measuring neural responses to 'uncanny valley' stimuli is that, without requiring conscious awareness or behavioural responses, they can facilitate detection of near-human stimuli. These types of near-human stimuli are becoming increasingly common in impersonation and identity evasion cases [1]. Simultaneously, we observe a growing market for reducing the uncanny valley effect for the benefit of android and robot integration. Exploring the potential of non-invasive brain recordings will benefit various applied fields as a result.

## Conclusion

We have identified neural correlates of the uncanny valley effect that are consistent across two experiments, using androids and hyper-realistic silicone masks. In both cases, perceptual discrimination from real human faces was possible, but more challenging than discriminating from mechanical robots or Halloween masks. Univariate differences in the ERP signal were unconvincing, but a more sensitive multivariate classification analysis identified differences at both early (100–200 ms) and later (around 600 ms) time points. These findings suggest the importance of both bottom up and top down influences on the subjective experience of the uncanny valley. Future work might extend these findings to more dynamic stimuli, and explore potential applications for improving android and avatar generation.

## Author contributions

**Conceptualization:** Shona Fitzpatrick, Ailish K. Byrne, Alex Headley, Jet G. Sanders, Helen Petrie, Rob Jenkins, Daniel H. Baker.

**Data curation:** Daniel H. Baker.

**Formal analysis:** Ailish K. Byrne, Daniel H. Baker.

**Funding acquisition:** Shona Fitzpatrick, Jet G. Sanders, Daniel H. Baker.

**Investigation:** Shona Fitzpatrick, Ailish K. Byrne, Alex Headley, Jet G. Sanders.

**Methodology:** Shona Fitzpatrick, Ailish K. Byrne, Alex Headley, Jet G. Sanders, Helen Petrie, Rob Jenkins, Daniel H. Baker.

**Project administration:** Helen Petrie, Rob Jenkins, Daniel H. Baker.

**Resources:** Shona Fitzpatrick, Ailish K. Byrne, Alex Headley, Jet G. Sanders, Rob Jenkins, Daniel H. Baker.

**Software:** Jet G. Sanders, Daniel H. Baker.

**Supervision:** Helen Petrie, Rob Jenkins, Daniel H. Baker.

**Visualization:** Daniel H. Baker.

**Writing – original draft:** Shona Fitzpatrick, Ailish K. Byrne, Alex Headley, Daniel H. Baker.

**Writing – review & editing:** Shona Fitzpatrick, Ailish K. Byrne, Alex Headley, Jet G. Sanders, Helen Petrie, Rob Jenkins.

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
