## [Decision Letter · Decision Letter 0]

9 Dec 2024

PONE-D-24-42223Neural correlates of the uncanny valley effect for robots and hyper-realistic masksPLOS ONE

Dear Dr. Baker,

Thank you for submitting your manuscript to PLOS ONE. After careful consideration, we feel that it has merit but does not fully meet PLOS ONE’s publication criteria as it currently stands. Therefore, we invite you to submit a revised version of the manuscript that addresses the points raised during the review process.

We look forward to receiving your revised manuscript.

Kind regards,

Rae Yule Kim

Academic Editor

PLOS ONE

Journal Requirements:

“SF was funded by a YorRobots Venables internship. JGS was funded by the Economic and Social Research

Council (Studentship ES/J500215/1) and Prins Bernhard Cultuurfonds (grant E/30/30.13.0630/HVH/IE).

Also supported by BBSRC grant BB/V007580/1 awarded to DHB.”

“SF was funded by a YorRobots Venables internship. JGS was funded by the Economic and Social Research Council (Studentship ES/J500215/1) and Prins Bernhard Cultuurfonds (grant E/30/30.13.0630/HVH/IE). Also supported by BBSRC grant BB/V007580/1 awarded to DHB.”

“SF was funded by a YorRobots Venables internship. JGS was funded by the Economic and Social Research

Council (Studentship ES/J500215/1) and Prins Bernhard Cultuurfonds (grant E/30/30.13.0630/HVH/IE).

Also supported by BBSRC grant BB/V007580/1 awarded to DHB.”

4. Please update your submission to use the PLOS LaTeX template. The template and more information on our requirements for LaTeX submissions can be found at http://journals.plos.org/plosone/s/latex.

5. We note that Figure 1 includes an image of a [patient / participant / in the study].

Additional Editor Comments (if provided):

The reviewers have expressed a positive view of your paper’s contribution to the field of human-robot interaction and are supportive of its acceptance, pending minor revisions.

In particular, the reviewers recommend enhancing the discussion with real-world examples to underscore the relevance of your findings. Additionally, they suggest providing further clarification on certain aspects of your methodology to improve the overall clarity of the paper.

Please address the reviewers’ comments and include a response letter for each reviewer with your revised manuscript.

We look forward to receiving your revised submission.

Reviewers' comments:

Reviewer's Responses to Questions

**Comments to the Author**

1. Is the manuscript technically sound, and do the data support the conclusions?

Reviewer #1: Yes

Reviewer #2: Partly

Reviewer #3: Yes

2. Has the statistical analysis been performed appropriately and rigorously? 

Reviewer #1: Yes

Reviewer #2: Yes

Reviewer #3: Yes

3. Have the authors made all data underlying the findings in their manuscript fully available?

Reviewer #1: Yes

Reviewer #2: Yes

Reviewer #3: Yes

4. Is the manuscript presented in an intelligible fashion and written in standard English?

Reviewer #1: Yes

Reviewer #2: Yes

Reviewer #3: Yes

5. Review Comments to the Author

Reviewer #1: Dear Authors,

I commend you on your fascinating study of the neural correlates of the uncanny valley effect. Your methodology is solid, and the findings provide valuable insights into human responses to artificial entities, with important implications for AI and robotics design.

Please find more detailed feedback and suggestions in the reviewer attachment. In summary, I recommend the following:

1. Simplify some technical terms in the abstract and introduction for broader accessibility.

2. Provide more context and explanation in the Methods section, especially regarding statistical analysis and EEG terms.

3. Expand figure captions and define EEG terms like P100 and N170 to aid readers’ understanding.

4. Consider exploring further applications of your findings in the Discussion and including alternative explanations.

5. In the Conclusion, suggest concrete directions for future research.

6. Enhance clarity by adding clearer labels in figures and defining technical terms earlier.

I believe your manuscript has great potential and, with these minor revisions, will be ready for publication.

Best regards,

Reviewer #2: The authors presented an interesting work on detecting the uncanny valley effect for robots and human-like subjects. The statistical analysis and results are well presented; however, the following changes can further improve the manuscript.

1. The motivation and application of the study should also be added to the introduction section.

2. The literature survey for similar works is short and needs improvement.

3. An image of the experimentation procedure should be added.

Reviewer #3: Genearl comments:

This study provides a valuable contribution to the understanding of the neural correlates of the uncanny valley effect through the use of EEG analysis on static images of androids and hyper-realistic silicone masks. The findings, such as the identification of distinct neural responses in early (100-200ms) and late (500-800ms) time windows, offer fresh insights into both bottom-up and top-down processes involved in this phenomenon. Additionally, the study demonstrates consistency across two experiments, enhancing the reliability of its conclusions.

However, there are areas that require further clarification and development to strengthen the manuscript. For example, the introduction does not fully articulate the motivation behind using static images and their advantage over dynamic stimuli in uncovering neural mechanisms. Additionally, the experimental design lacks detail regarding stimulus selection criteria and the rationale behind conditions. While EEG and behavioral data were analyzed, their integration remains insufficient, and the neural mechanisms underlying the temporal windows are not explored in depth. Furthermore, subjective ratings (e.g., uncanniness and realism) are underutilized, as their relationship with neural or behavioral data is not analyzed. Lastly, while the discussion highlights potential applications, such as improving AI-generated images, it does not provide sufficient details on how these applications might be implemented.

Major Issues and Recommendations

1. Insufficient Expression of Research Motivation

The introduction does not clearly articulate the scientific significance of the study, such as why investigating static images can complement existing research on dynamic stimuli. This may lead to ambiguity regarding the study’s uniqueness and contributions.

Recommendation: Emphasize the importance of static image research: “Although existing studies focus on dynamic stimuli, static images allow for a more precise investigation of the neural mechanisms underlying the uncanny valley effect, particularly by eliminating motion-related confounds.”

2. Lack of Clarity in Experimental Objectives

The description of the experimental design and objectives is too vague, failing to specify the criteria for selecting static images or the distinctions between experimental conditions.

Provide a clearer explanation of the study’s goals and design, detailing the selection process for images and the specific distinctions between conditions.

3. Insufficient Discussion of EEG Data Analysis

The ERP differences are only briefly discussed, without exploring the potential neural mechanisms of specific time windows (e.g., whether the 300-400ms differences are linked to cognitive conflict).

Deepen the discussion on ERP mechanisms: “The 300-400ms time window may reflect cognitive processing stages, such as determining whether a stimulus conforms to categorical expectations. Differences between human and robot conditions could indicate the unique processing advantage of human faces.”

4. Weak Integration of Results

There is no clear explanation of how d′ values, reaction times, and EEG data integrate to address the research question, making the relationship between methodologies unclear.

Connect different data types to answer the research question: “By integrating d′ values, reaction times, and EEG data, we explore how neural activity in specific time windows correlates with behavioral metrics like accuracy and response time.”

5. Poor Integration of Subjective Ratings with Neural Data

Perception ratings (e.g., uncanniness, realism) are not linked to EEG data. The connection between subjective experiences and neural signals is critical but underexplored.

Analyze the relationship between ratings and EEG data: “Higher uncanniness ratings may correspond to greater ERP differences in the late time window (600ms), indicating a link between subjective perceptions and neural responses.”

Minor Issues and Recommendations

1. Weak Literature Review

The discussion of existing research is underdeveloped, particularly regarding theoretical disagreements on the neural mechanisms (bottom-up vs. top-down processes).

Expand the literature review: “The root of the uncanny valley effect remains debated: does it arise primarily from bottom-up sensory conflicts, or from higher-level cognitive processes? Resolving this question is critical to understanding its fundamental mechanisms.”

2. Lack of Explanation for Stimulus Presentation Time

The rationale behind the stimulus presentation times (500ms for Experiment 1, 250ms for Experiment 2) and random blank periods is not explained.

Justify the choices: “The 500ms and 250ms durations were chosen to provide sufficient information for judgment while avoiding task fatigue. Randomized blank periods were designed to reduce carryover effects and prevent anticipatory biases.”

3. Insufficient Analysis of Image Background Differences

Experiment 1 used gray backgrounds, while Experiment 2 used natural backgrounds, but the potential impact of this difference is not addressed.

Discuss the impact of background differences: “While the backgrounds differed across experiments, we hypothesize that the primary task was not affected, as participants focused on the foreground stimuli.”

4. Lack of Detail in d′ Value Calculation

The calculation method for d′ values lacks detail, particularly for edge cases (e.g., hit rate = 1 or 0) and the rationale for capping infinite values.

Provide detailed explanations: “d′ values were capped at 5 to prevent outliers from skewing the results, following established conventions in signal detection theory.”

5. Insufficient Interpretation of Figures

Figures (e.g., 2 and 3) are not adequately interpreted, leaving the reader unclear about their implications.

Offer more detailed interpretations: “Figure 2a shows that robot stimuli have significantly higher d′ values than humanoids and humans, emphasizing the role of visual salience.”

6. Limited Context for Multivariate Pattern Analysis

High classification accuracy for human vs. robot stimuli (72%) and lower accuracy for humanoids (62%) is presented without explaining the reasons for this disparity, such as the salience of visual features.

Explore potential causes for classification differences: “The high classification accuracy for human vs. robot stimuli may be attributed to the salient mechanical elements of robots, whereas the lower accuracy for humanoids reflects their ambiguous human-like appearance, leading to confusion.”

7. Lack of Discussion on Cross-Experiment Consistency

While the consistency of findings across experiments is mentioned, its implications for the uncanny valley theory are not explored.

Emphasize the theoretical significance of consistent results: “The consistency of results across humanoids and hyper-realistic silicone masks suggests that the neural characteristics of the uncanny valley effect are generalizable across stimulus categories, indicating its broad applicability.”

8. Insufficient Analysis of Temporal Windows

The mechanisms underlying the two temporal windows (100ms and 500-800ms) are oversimplified, with no mention of their potential links to emotion processing or visual memory.

Provide detailed hypotheses for each window:

100ms window: “This window likely reflects rapid processing of low-level visual features, such as edges or color contrasts, consistent with P100 and N170 components.”

500-800ms window: “This window may involve higher-level cognitive processes, such as evaluating emotional content or judging authenticity.”

The study makes a significant contribution by identifying neural correlates of the uncanny valley effect and providing practical implications for AI and identity detection technologies. However, by addressing the above suggestions, the authors can further enhance the clarity, depth, and impact of their work.

6. PLOS authors have the option to publish the peer review history of their article (what does this mean?). If published, this will include your full peer review and any attached files.

Reviewer #1: **Yes: **Osamah AL-Qalisi

Reviewer #2: No

Reviewer #3: No

---

## [Decision Letter · Decision Letter 1]

7 Feb 2025

Neural correlates of the uncanny valley effect for robots and hyper-realistic masks

PONE-D-24-42223R1

Dear Dr. Baker,

We’re pleased to inform you that your manuscript has been judged scientifically suitable for publication and will be formally accepted for publication once it meets all outstanding technical requirements.

Kind regards,

Rae Yule Kim

Academic Editor

PLOS ONE

Reviewers' comments:

Reviewer's Responses to Questions

**Comments to the Author**

1. If the authors have adequately addressed your comments raised in a previous round of review and you feel that this manuscript is now acceptable for publication, you may indicate that here to bypass the “Comments to the Author” section, enter your conflict of interest statement in the “Confidential to Editor” section, and submit your "Accept" recommendation.

Reviewer #2: All comments have been addressed

Reviewer #3: All comments have been addressed

2. Is the manuscript technically sound, and do the data support the conclusions?

Reviewer #2: Yes

Reviewer #3: Yes

3. Has the statistical analysis been performed appropriately and rigorously? 

Reviewer #2: Yes

Reviewer #3: Yes

4. Have the authors made all data underlying the findings in their manuscript fully available?

Reviewer #2: Yes

Reviewer #3: Yes

5. Is the manuscript presented in an intelligible fashion and written in standard English?

Reviewer #2: Yes

Reviewer #3: Yes

6. Review Comments to the Author

Reviewer #2: (No Response)

Reviewer #3: The revised manuscript has significantly improved, addressing previous concerns effectively. The clarity, analysis, and overall quality are now strong. I recommend acceptance in its current form. Well done to the authors.

7. PLOS authors have the option to publish the peer review history of their article (what does this mean?). If published, this will include your full peer review and any attached files.

Reviewer #2: No

Reviewer #3: No

---

## [Editor Report · Acceptance letter]

PONE-D-24-42223R1

PLOS ONE

Dear Dr. Baker,

I'm pleased to inform you that your manuscript has been deemed suitable for publication in PLOS ONE. Congratulations! Your manuscript is now being handed over to our production team.

Kind regards,

on behalf of

Dr. Rae Yule Kim

Academic Editor

PLOS ONE